# The Construction and Immunogenicity Analyses of a Recombinant Pseudorabies Virus with Senecavirus A VP2 Protein Coexpression

Qian Tao,[a] Ling Zhu,[a,b] Lei Xu,[a] Yanting Yang,[a] Yang Zhang,[a] Zheyan Liu,[a] Tong Xu,[a] Jianhua Wen,[a] Lishuang Deng,[a] Yuancheng Zhou,[c] ⓘZhiwen Xu[a,b]

aCollege of Veterinary Medicine, Sichuan Agricultural University, Chengdu, China
bKey Laboratory of Animal Disease and Human Health of Sichuan Province, Chengdu, China
cAnimal Breeding and Genetics Key Laboratory of Sichuan Province, Sichuan Animal Science Academy, Chengdu, China

Qian Tao and Ling Zhu contributed equally to this work. The author order was determined based on contributions made and common consultation.

**ABSTRACT**   Senecavirus A (SVA)-associated porcine idiopathic vesicular disease (PIVD) and pseudorabies (PR) are highly contagious swine diseases that pose a significant threat to the swine industry in China. Since there is currently no effective commercial vaccine against SVA, the virus has spread widely throughout China and its pathogenicity has increased over the last decade. In this study, a recombinant strain named rPRV-XJ-ΔTK/gE/gI-VP2 was constructed by using the pseudorabies virus (PRV) variant strain XJ as the parental virus and by deleting the TK/gE/gI gene while coexpressing SVA VP2. The recombinant strain can stably proliferate and express foreign protein VP2 in BHK-21 cells while having a similar virion appearance to that of the parental strain. rPRV-XJ-ΔTK/gE/gI-VP2 is safe and effective for BALB/c mice, inducing high levels of neutralizing antibodies against both PRV and SVA, providing 100% protection from the virulent PRV strain. Histopathological examination and quantitative PCR (qPCR) assay have demonstrated that SVA can infect mice through intranasal inoculation, while the vaccination of mice with rPRV-XJ-ΔTK/gE/gI-VP2 can significantly reduce SVA virus copies and alleviate the pathological inflammatory changes in the heart and liver. The evaluation of the safety and immunogenicity indicates that rPRV-XJ-ΔTK/gE/gI-VP2 holds promise as a candidate vaccine against PRV and SVA.

**IMPORTANCE**   This study reports the construction of a recombinant PRV with SVA for the first time, and the resulting virus, rPRV-XJ-ΔTK/gE/gI-VP2, can induce high levels of neutralizing antibodies against both PRV and SVA in model mice. These findings provide valuable insights for evaluating the effectiveness of rPRV-XJ-ΔTK/gE/gI-VP2 as a vaccine for pigs. Additionally, this study reports transient SVA infection in mice, with qPCR assays showing that the copies of the SVA 3D gene peaked at 3 to 6 days postinfection and fell below the sensitivity threshold by 14 days postinfection. The copies of the gene were more regular and at a higher level in the heart, liver, spleen, and lung tissue.

**KEYWORDS**   SVA, mice, VP2, PRV, multivalent vaccines

Senecavirus A (SVA), formerly known as Seneca Valley virus (SVV), is the only member of the genus *Senecavirus*, family *Picornaviridae*, with a diameter of 17 to 25 nm (1, 2). The virus genome is a nonenveloped positive single-stranded RNA of approximately 7.2 kb in length, with the genome organization of 5′-untranslated region (UTR)-VP4-VP2 -VP3-VP1-2A-2B-2C-3A-3B-3C-3D-3′-UTR (3). SVA was first discovered as a contaminant of the PER.C6 cell line used for cultivating viral vectors in 2002 and was named SVA-001 (4). The first case of SVA-associated porcine idiopathic vesicular disease (PIVD) in pigs was reported in Canada in June 2007, with the virus being found in lesions (5). Subsequently, SVA was detected in

Address correspondence to Zhiwen Xu, abtcxzw@126.com.

The authors declare no conflict of interest.

the United States in 2012 (6), and in 2014 and 2015, outbreaks of SVA-associated PIVD in sows and SVA-associated epidemic transient neonatal losses (ETNL) occurred in Brazil and the United States, respectively (7, 8). Since then, outbreaks of SVA-associated PIVD and ETNL have been reported in several other countries, including China (8), Thailand (9), Vietnam (10), Colombia (11), and Chile (12). Phylogenetic analysis indicates the existence of three clades of the virus, with clade I containing the historical strain SVA-001, clade II containing U.S. SVA strains identified between 1988 and 1997, and clade III containing global SVA strains identified after 2001 (13). Typical clinical symptoms of SVA infection are vesicular lesions on the snout and feet of nursing and gestating sows, which are similar to those caused by foot-and-mouth disease virus (FMDV) (13).

SVA has the following four structural proteins: VP1, VP2, VP3, and VP4. The report on the structure and function of VP2 provides valuable information for distinguishing SVA from other picornaviruses. The structure of VP2 may also determine the cell tropism of SVA (2). Unlike most picornaviruses, neutralizing epitopes of SVA may be contained in VP2 rather than in VP1 (14).

Pseudorabies virus (PRV), an alphaherpesvirus with double-stranded DNA of approximately 143 kb, contains almost 70 open reading frames (ORFs) that encode 70 to 100 viral proteins consisting of structural, nonstructural, and virulence-associated protein (15). Several virulence-associated proteins, such as the TK, gE, gI, and gG genes, can be deleted or replaced by foreign genes without affecting PRV replication or the immune response (16). Due to its high capacity, PRV has become a powerful vector system for expressing exogenous proteins. It can express the main immunogenic protein of other pathogens, such as the Cap protein of porcine circovirus (PCV) (17), VP2 of porcine parvovirus (PPV) (18), CD2v of African swine fever virus (ASFV) (19), S of porcine deltacoronavirus (PDCoV) (20), and GP5 and M of porcine reproductive and respiratory syndrome virus (PRRSV) (21).

In this study, we utilized homologous recombination and the CRISPR-Cas9 system to construct a recombinant PRV strain that expresses a functional SVA VP2 protein, named rPRV-XJ-ΔTK/gE/gI-VP2. We evaluated the safety of this strain and its ability to elicit humoral and cellular immune responses in mice. Our findings provide valuable evidence for future vaccine development against both SVA and PRV strains.

## RESULTS

**Construction of the recombinant virus rPRV-XJ-ΔTK/gE/gI-VP2.** The recombinant virus rPRV-XJ-ΔTK/gE/gI-VP2, which expresses the SVA VP2 protein, was constructed using CRISPR-Cas9 and homologous recombination technology (Fig. 1A). The SVA VP2 protein with an enhanced green fluorescent protein (EGFP) tag expression cassette was inserted into RRV-XJ-ΔTK and driven by the cytomegalovirus (CMV) endogenous promoter. The protein was expressed independently using the ribosomal jump translation of a 2A cleavage peptide. Green fluorescence recombinant viruses were observed in BHK-21 cells after plasmid transfection (Fig. 1B) and PRV-XJ-ΔTK infection (Fig. 1C). After three rounds of purification, the rPRV-XJ-ΔTK/gE/gI-VP2 strain was generated (Fig. 1E) and identified by PCR. The VP2 gene amplification was positive in rPRV-XJ-ΔTK/gE/gI-VP2 (Fig. 1D).

**Characterization of recombinant virus rPRV-XJ-ΔTK/gE/gI-VP2.** The rPRV-XJ-ΔTK/gE/gI-VP2 strain was propagated for 21 generations, and PCR was performed to detect the presence of VP2 gene amplification in the F5, F10, F15, and F21 generations. Compared with PRV-XJ, the VP2 gene amplification was positive, gE and gI genes were negative, and the TK gene was shortened due to the CRISPR-Cas9 cut (Fig. 2A). SVA-VP2 protein expression was examined in BHK-21 cells infected with rPRV-XJ-ΔTK/gE/gI-VP2 by Western blot and immunofluorescence assay (IFA). Compared with PRV-XJ, the band recognized by mouse anti-VP2 polyclonal antibody and the color developed by the horseradish peroxidase (HRP) reagent were specific to the sample infected with rPRV-XJ-ΔTK/gE/gI-VP2 (Fig. 2B). The expression of gB was detected in both the rPRV-XJ-ΔTK/gE/gI-VP2 and PRV-XJ groups. Furthermore, the BHK-21 cells infected with rPRV-XJ-ΔTK/gE/gI-VP2 showed positive immunofluorescence for the VP2 protein by the mouse anti-VP2 polyclonal antibody and color developed by the immunofluorescence secondary antibody (Fig. 2C).

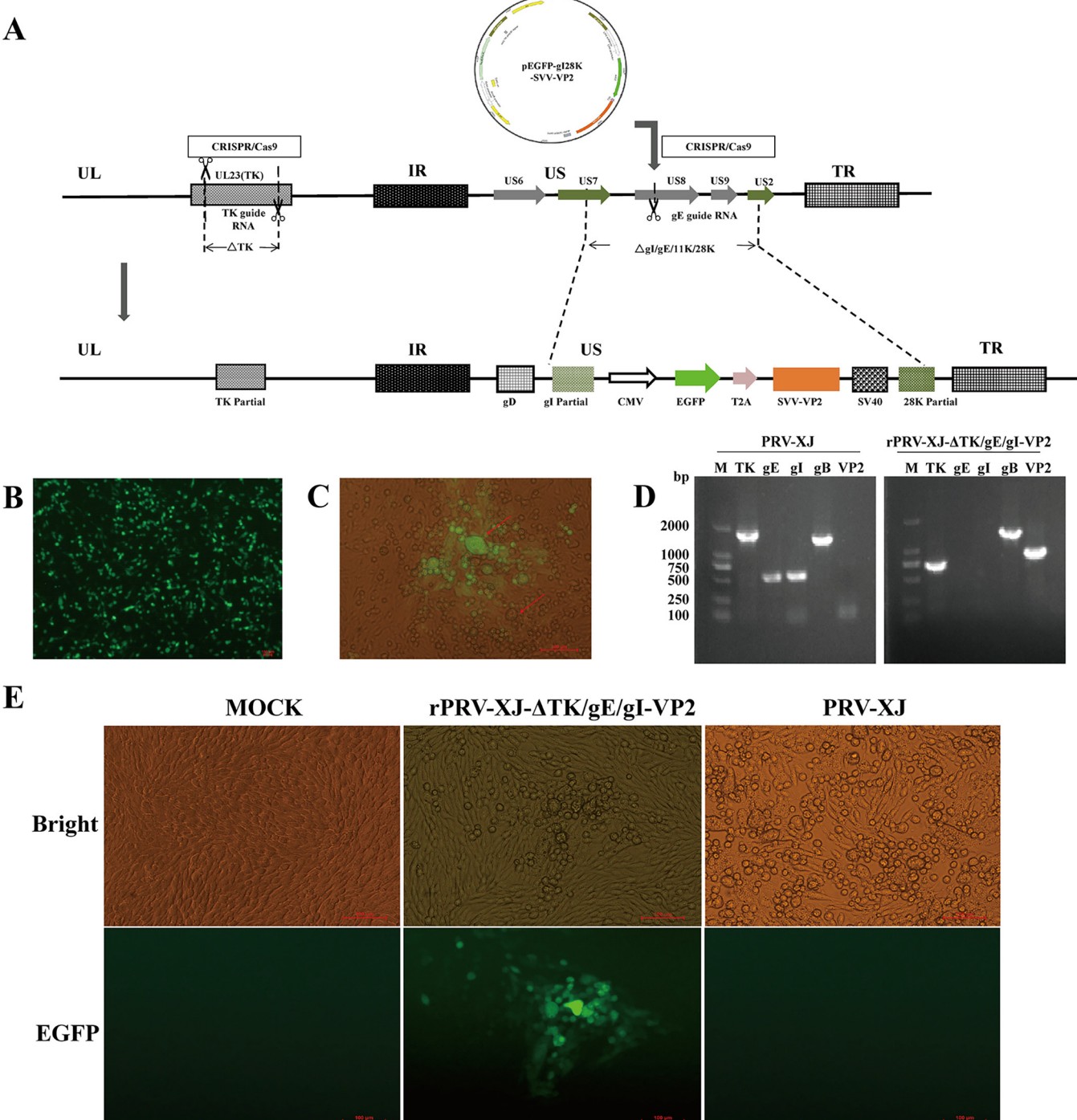

**FIG 1** Construction of the recombinant virus rPRV-XJ-ΔTK/gE/gl-VP2. (A) Schematic diagram of the recombinant pseudorabies virus (PRV) development strategy. (B) Fluorogram of eukaryotic transfer vector pEGFP-gl28K-SVV-VP2 transfection. (C) Plaque purification for the recombinant virus. Virus plaques are indicated with red arrows. Green plaques are the BHK-21 cells infected with recombinant virus. (D) PCR identification of the recombinant virus purification. Genes of gE and gl were replaced by SVA VP2, and the TK gene got shorter with the recombinant virus. (E) The BHK-21 cells infected with the recombinant strain and parental strain were observed under a fluorescence microscope.

To assess the impact of foreign fragment insertion, the Reed-Muench method was used to analyze one-step growth curves of PRV-XJ, rPRV-XJ-ΔTK/gE/gl, and rPRV-XJ-ΔTK/gE/gl-VP2, and the virus plaque generation capacity was analyzed by plaque experiments. The growth dynamic of rPRV-XJ-ΔTK/gE/gl-VP2 was similar to that of PRV-XJ and rPRV-XJ-ΔTK/gE/gl (Fig. 2D). The virus plaque generation capacity of rPRV-XJ-ΔTK/gE/gl-VP2 was similar to that of rPRV-XJ-ΔTK/gE/gl but lower than that of PRV-XJ (Fig. 2E). Transmission electron

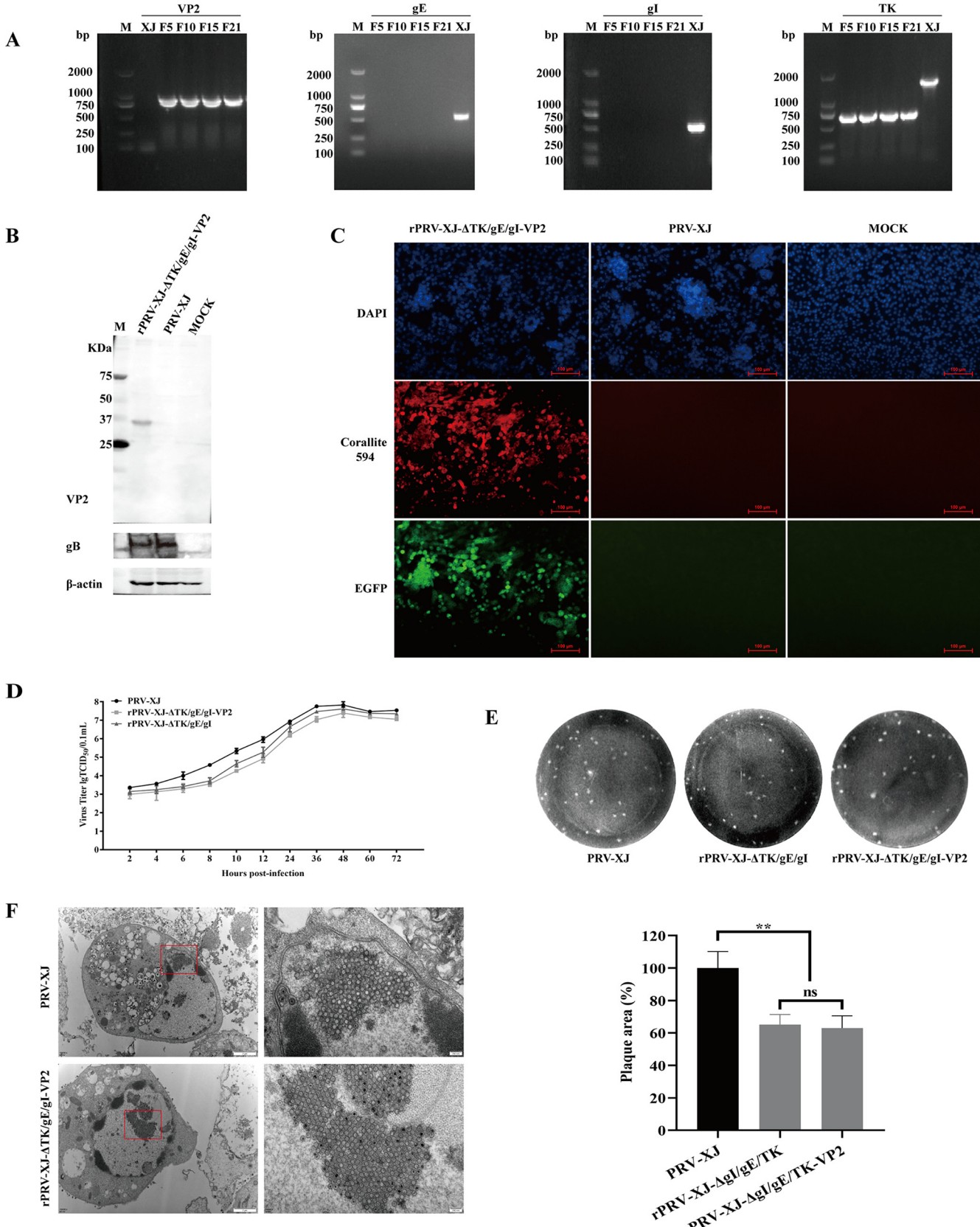

**FIG 2** Characterization of recombinant virus rPRV-XJ-ΔTK/gE/gI-VP2. (A) PCR identification of the deletion of TK, gE, and gI and the insertion of VP2 in recombinant virus (F5, F10, F15, and F21), with the parental strain PRV-XJ (XJ) as a control. (B) Western blotting of SVA VP2 protein in BHK-21 cells infected

microscopy was used to observe the rPRV-XJ-ΔTK/gE/gI-VP2 virion, which was similar to rPRV-XJ-ΔTK/gE/gI (21), and no significant difference was observed compared with PRV-XJ (Fig. 2F).

**rPRV-XJ-ΔTK/gE/gI-VP2 is safe for mice.** Mice were used to evaluate the safety of the recombinant virus. After 3 days post-PRV challenge (dpc), mice in the PRV-XJ group showed itching and scratching symptoms and subsequently died within a week. In contrast, mice in the rPRV-XJ-ΔTK/gE/gI-VP2 group inoculated with $10^5$, $10^6$, and $10^7$ 50% tissue culture infective dose ($TCID_{50}$) showed no clinical symptoms and survived until the end of the observation period (Fig. 3A). The shedding of the rPRV-XJ-ΔTK/gE/gI-VP2 virus in the mouse model was evaluated using qPCR. Following single footpad inoculation with $10^7$ $TCID_{50}$ of rPRV-XJ-ΔTK/gE/gI-VP2, the PRV IE180 gene copies peaked at 7 days postvaccination (dpv), and fell below the sensitivity threshold by 15 dpv (Fig. 3B). To evaluate the inflammatory reaction caused by the virus, levels of the proinflammatory cytokines interleukin-6 (IL-6) and tumor necrosis factor alpha (TNF-$\alpha$) were measured by enzyme-linked immunosorbent assay (ELISA) at 3 dpc. While high levels of IL-6 and TNF-$\alpha$ were induced by PRV-XJ infection, inoculation with each dose of recombinant strains did not cause upregulation of IL-6 (Fig. 3C) and TNF-$\alpha$ (Fig. 3D). Pathological changes in the brain tissues of mice in each group were analyzed by hematoxylin and eosin (H&E) staining. PRV-XJ infection caused neurological damage, including vacuolar neuronal degeneration, neuron phagocytosis, and nuclear cleavage, whereas no histopathological changes were observed in the recombinant strain group (Fig. 3E).

**rPRV-XJ-ΔTK/gE/gI-VP2 can induce the gB- and VP2-specific antibody and the PRV- and SVA-neutralizing antibody.** Figure 4A depicts the process of the mouse immunization experiment. Both anti-PRV- and anti-VP2-specific antibodies produced by the immunized mice were tested by ELISA. The PRV gB-specific antibody was examined by blocking ELISA, and the SVA-VP2-specific antibody was examined by the ELISA established by our lab. Compared with the Dulbecco's modified Eagle medium (DMEM) group, the gB-specific antibody increased significantly and turned positive after 21 dpv (7 dpv for the booster vaccination) in both the rPRV-XJ-ΔTK/gE/gI- and rPRV-XJ-ΔTK/gE/gI-VP2-vaccinated groups (Fig. 4B). Compared with the DMEM group, the gB-specific antibody significantly increased and turned positive after 21 days post vaccination (dpv) (7 dpv for the booster vaccination) in both rPRV-XJ-ΔTK/gE/gI- and rPRV-XJ-ΔTK/gE/gI-VP2-vaccinated groups (Fig. 4B). In the rPRV-XJ-ΔTK/gE/gI-VP2-vaccinated group, the SVA-VP2-specific antibody significantly increased at the same time and maintained a high level that was not detected in the rPRV-XJ-ΔTK/gE/gI group (Fig. 4C).

PRV- and SVA-neutralizing antibodies were tested by plaque reduction assay. These antibodies were produced from 7 dpv and reached a maximum at 28 to 35 dpv. The highest PRV-neutralizing antibody titers were $2^{5.08}$ (Fig. 4D), and the highest SVA-neutralizing antibody titers was $2^{4.42}$ (Fig. 4E) in the rPRV-XJ-ΔTK/gE/gI-VP2-vaccinated group. High-level PRV-neutralizing antibodies were also detected in the rPRV-XJ-ΔTK/gE/gI group, and the growth trend of the titer curve was similar to that of the rPRV-XJ-ΔTK/gE/gI-VP2-vaccinated group (Fig. 4C). As a negative control, mice injected with DMEM did not produce any neutralizing antibodies of PRV and SVA.

**rPRV-XJ-ΔTK/gE/gI-VP2 can induce cellular immune responses in a mouse model.** At 14 dpv, flow cytometry was used to detect T cell types. The results showed a significant increase in the proportion of $CD3^+$ T cells in murine splenic lymphocytes after rPRV-XJ-ΔTK/gE/gI or rPRV-XJ-ΔTK/gE/gI-VP2 vaccination. And both the gene deletion strain and recombinant strain infections significantly increased the proliferation of $CD3^+$ $CD4^+$ T cells (T helper [Th] cells) and $CD3^+$ $CD8^+$ T cells (cytotoxic T cells [CTLs]) (Fig. 5A). Interferon-$\gamma$ (IFN-$\gamma$) can serve as a criterion for evaluating the specific T cell-mediated immune effects of vaccines (22). The level of IFN-$\gamma$ significantly increased in both the rPRV-XJ-ΔTK/gE/gI and rPRV-XJ-

**FIG 2** Legend (Continued)
with the recombinant viruses. Mock, BHK-21 cell lysis products. (C) Immunofluorescence assay (IFA) detection of VP2 protein expression. Red, VP2 protein bound with anti-VP2 antibody and labeled by Corallite 594; green, EGFP tag in recombinant virus; blue, 4',6-diamidino-2-phenylindole (DAPI)-stained BHK-21 cell nucleus. (D) One-step growth curve of the recombinant strain and parent strain in BHK-21. (E) Comparison of the plaque size of BHK-21 cells infected with the recombinant strain and the parent strain. (F) Transmission electron microscopy analysis of BHK-21 cells infected with the recombinant strain and the parent strain. Data are presented as mean $\pm$ SD ($n = 3$); ns, not significant; **, $P < 0.01$.

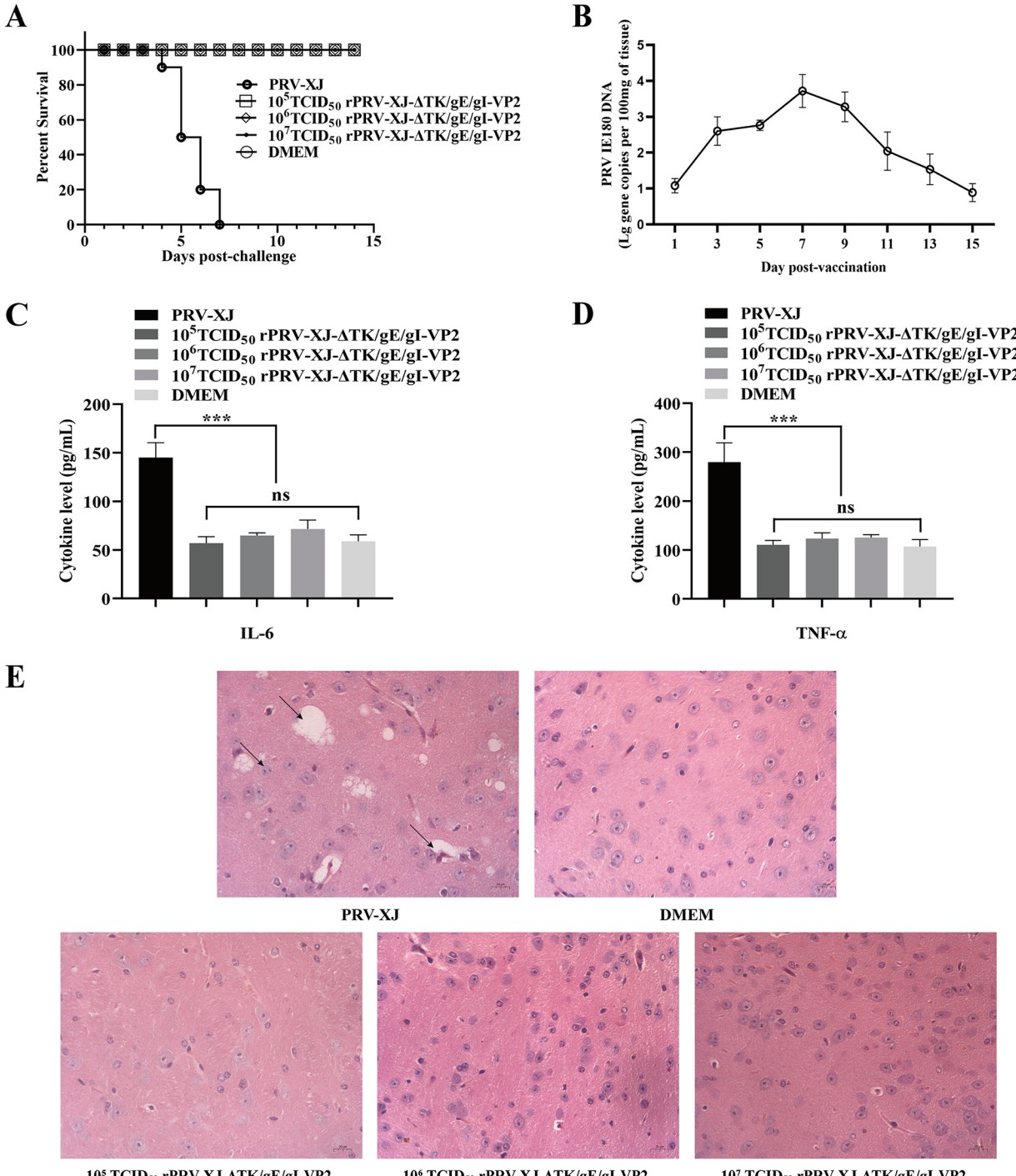

**FIG 3** The safety assessment of the recombinant virus. (A) Survival curves of mice infected with the parent strain and different titers of the recombinant strain. (B) The serum IL-6 was detected by ELISA. (C) The serum TNF-α was observed by ELISA. (D) Pathological observation of H&E staining in brain tissue of mice infected with the parent strain, DMEM, or different titers of the recombinant strain. The arrows on the left to the right show the nuclear cleavage, vacuolate neuronal degeneration, neuron phagocytosis. Data are presented as mean ± SD ($n = 3$); ***, $P < 0.005$.

ΔTK/gE/gI-VP2 groups compared with that of the DMEM group (Fig. 5B), with no significant difference observed between the two vaccinated groups.

The splenic lymphocytes of mice were stimulated by ConA or inactivated SVA to investigate cell proliferative activity using a CCK8 kit at 14 and 28 dpv. Compared with the DMEM group, the stimulation index (SI) of rPRV-XJ-ΔTK/gE/gI- and rPRV-XJ-ΔTK/gE/gI-VP2-vaccinated groups was significantly improved under ConA stimulation. ConA is a phytohemagglutinin

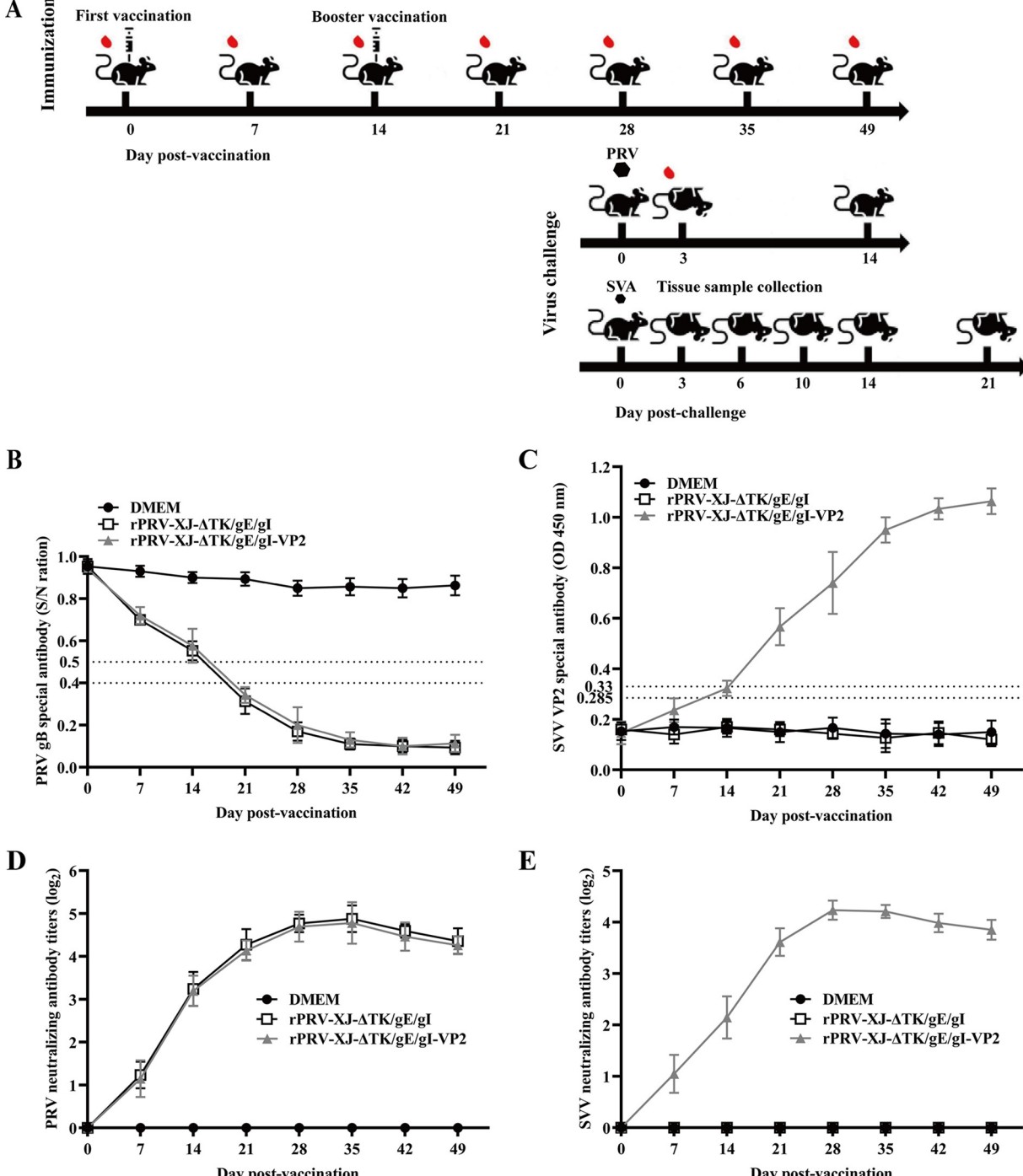

**FIG 4** Immune responses after intranasal immunization with recombinant virus. (A) Graphical representation of immunization/virus challenge experiment. (B) The levels of PRV gB-specific antibodies in the serum from 7 dpv to 49 dpv after vaccination were measured by blocking ELISA. (C) SVA VP2-specific antibodies were measured by ELISA that our lab established at indicated time points. (D) Neutralizing antibody titers against the PRV-XJ strain. (E) Neutralizing antibody titers against the SVA strain.

with potent mitogenic capacity, which can be used as a positive standard for stimulating splenic lymphocyte proliferation (23). While under inactivated SVA stimulation, the SI value was improved only in the recombinant strain group (Fig. 5C).

The levels of IL-2 and IL-4 were notably higher in both vaccinated groups than those in the DMEM group (Fig. 5D). The levels of IL-2 were similar between the two vaccinated groups, while the level of IL-4 in the rPRV-XJ-ΔTK/gE/gI-VP2 group was slightly higher than that of the rPRV-XJ-ΔTK/gE/gI group.

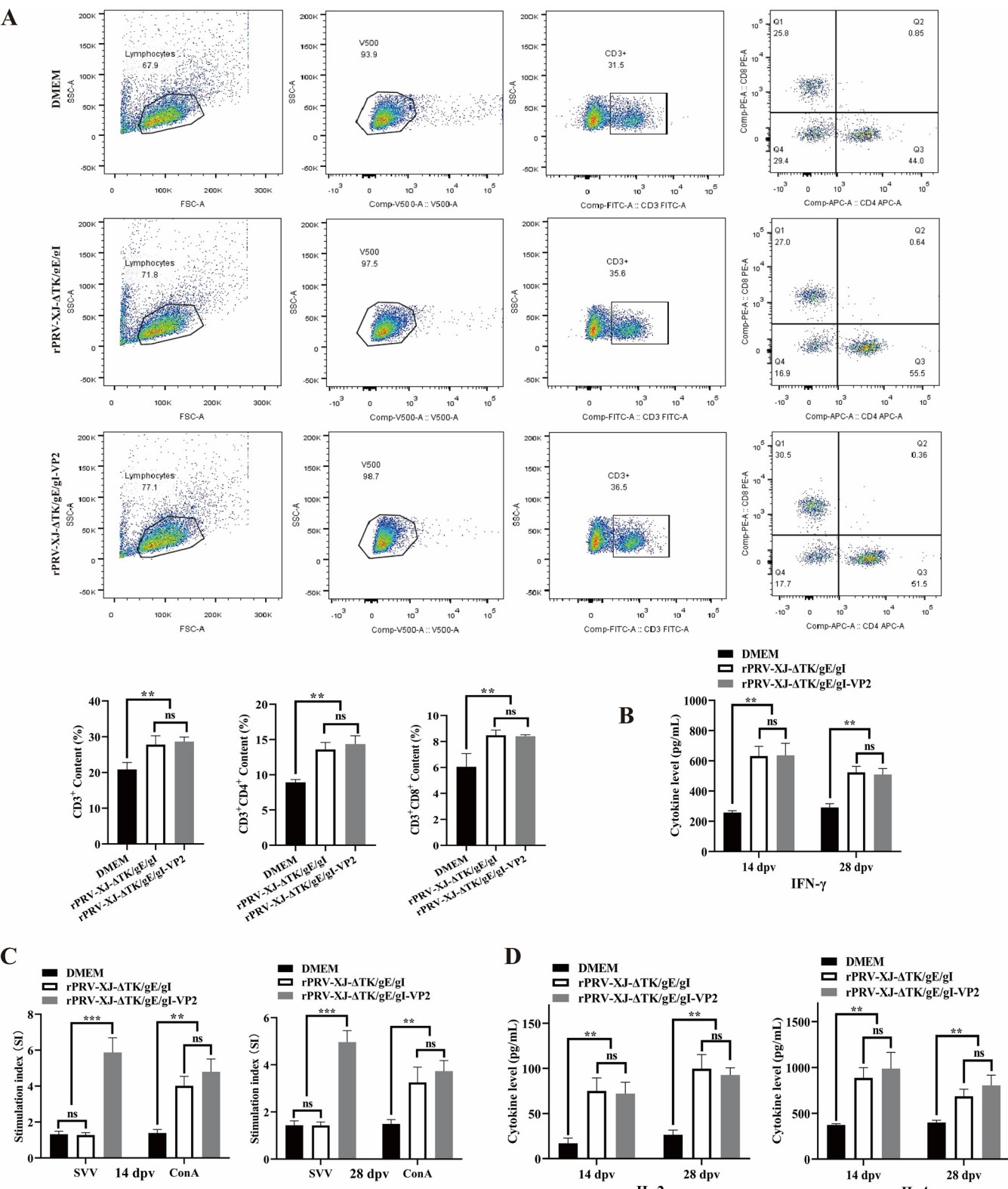

FIG 5 Cell immune responses analyses of rPRV-XJ-ΔTK/gE/gI, rPRV-XJ-ΔTK/gE/gI-VP2, and DMEM groups. (A) Representative gating strategy for T cells and the flow cytometry analysis result of CD3+, CD3+ CD4+, and CD3+ CD8+ T cell populations at 14 dpv after vaccination. (B) The serum IFN-γ was detected by ELISA. (C) Splenic lymphocyte-specific proliferative responses assay. (D) The serum IL-2 and IL-4 were detected by ELISA. Data are presented as mean ± SD (n = 3); **, P < 0.01; ***, P < 0.005.

**rPRV-XJ-ΔTK/gE/gI-VP2 can provide 100% protection to mice from challenge by the PRV virulent strain and reduce the tissue load and pathological injury of SVA in mice.** After 28 days postvaccination, 200 $\mu$L $10^5$ TCID$_{50}$ of the wild-type PRV-XJ strain was injected into the footpad of mice of each group, and the mice were observed for 14 dpc. Mice in the DMEM group showed obvious clinical symptoms of PRV, such as hyperkinesia, ruffled fur, pruritus, and hunching, and all mice died within 6 days postchallenge. In contrast, mice in the rPRV-XJ-ΔTK/gE/gI and rPRV-XJ-ΔTK/gE/gI-VP2 groups did not show such clinical symptoms of PRV and survived until the end of the observation period (Fig. 6A). Histopathological examinations of mouse brains were performed, and the mice of these vaccinated groups did not show significant histopathological changes in the tissue, while vacuolar neuronal degeneration, neuron phagocytosis, and nuclear cleavage of the brains were observed in the DMEM group mice (Fig. 6B). The cytokine levels were tested to evaluate the inflammatory reaction, and there was a highly significant upregulation of IL-6 and TNF-$\alpha$ in the DMEM group compared with those in the two vaccinated groups at 3 days postchallenge with PRV-XJ (Fig. 6C).

To simulate oronasal transmission of SVA in pigs, mice in each group were injected with 50 $\mu$L $10^5$ TCID$_{50}$ of SVA via the nose. After SVA infection, mice in each group did not show obvious clinical symptoms or death. The gene copy numbers of SVA in mouse tissues of each group were detected to evaluate the protection of rPRV-XJ-ΔTK/gE/gI-VP2 on SVA-infected mice. The 3D gene is a conserved gene of SVA (24), and in this study, it was used as the template to establish the qPCR standard curve for SVA detection ($y = -3.4653x + 36.194$, $R^2 = 0.9993$). The results revealed that SVA infection in mice was transient. In the DMEM group, gene copies of SVA in the heart, liver, spleen, lung, mesenteric lymph nodes, and feces were higher than those in other tissues and peaked at 3 to 6 dpc. At 14 dpc, the gene copies of SVA 3D decreased below the sensitivity threshold of the assay (Fig. 6D). Compared with the DMEM group, the gene copies in these tissues were significantly reduced and dropped below the threshold at 10 dpc (Fig. 6E) in mice vaccinated with rPRV-XJ-ΔTK/gE/gI-VP2.

Histopathological examination of these tissues was performed at 6 dpc. In the DMEM group, significant endocardial damage was observed, with the heart cavity containing numerous red blood cells (RBCs) and plasma and a large number of granulocytes around the RBCs. Myocardial cells showed slight swelling and loose cytoplasm. Hepatocytes showed extensive swelling, balloon-like degeneration, and cytoplasm vacuolation. Hepatic vessels were dilated and filled with RBCs and granulocytes. However, other tissues did not show significant pathological changes. In contrast, milder symptoms were observed in the rPRV-XJ-ΔTK/gE/gI-VP2 group. There was no obvious endocardial injury in the heart, but a few RBCs leaked, and liver cells showed slightly swollen and loose cytoplasm (Fig. 6F). Figure 4A depicts the process of the viral challenge experiment.

## DISCUSSION

SVA is a new type virus for the pig industry in China. The first outbreak of SVA occurred in Guangdong Province in March 2015, and it subsequently spread to other provinces, causing serious economic losses (25). Two subbranches of SVA have been isolated and identified in China. The SVA strains isolated before 2016 had high nucleotide homology with Canadian (26) and Brazilian strains (27), while the strains isolated after 2016 were more closely to United States strains (28). In addition to vesicular lesions, SVA infection also causes miscarriage in sows and acute death of newborn piglets (29). Since there is no effective commercial vaccine against SVA, and the virus can be transmitted through carrier pigs and pregnant sows, SVA represents a hidden danger to the pig industry (30).

Two studies have been reported to evaluate the protective effect of SVA inactivated vaccine in finishing pigs (31, 32). However, this type of vaccine has limitations, such as time-consuming and scale-restricted preparation, loss of virus infectivity, and immunity failure caused by maternal antibodies. In contrast, live vector vaccines do not have these drawbacks. Moreover, VP1 and VP3 are involved in SVA-induced autophagy by activating the AMPK-MAPK-MTOR signaling pathway (33), so avoiding immune evasion of these proteins can improve vaccine protection. Gene deletion-attenuated PRV is a mature live

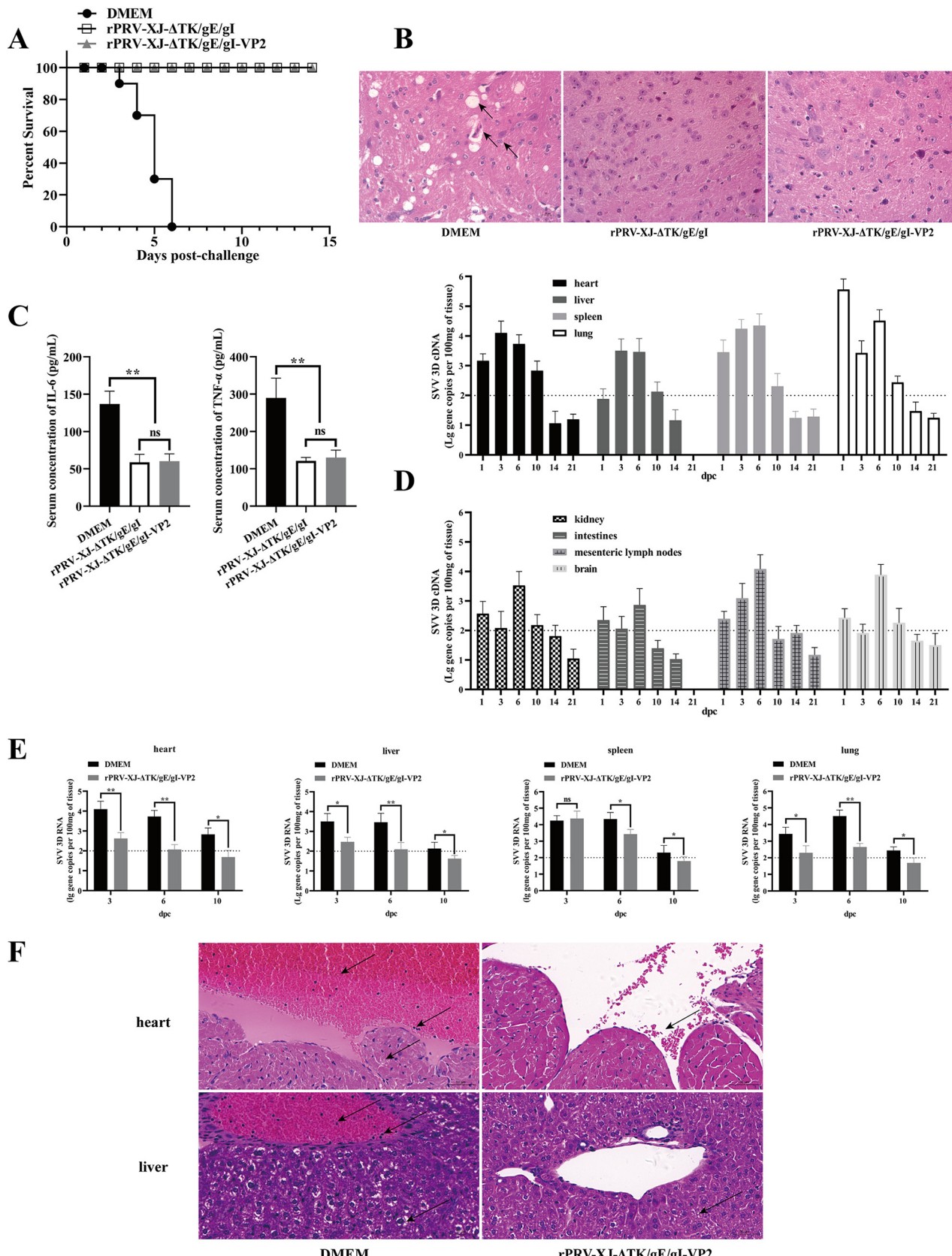

**FIG 6** rPRV-XJ-ΔTK/gE/gI-VP2 help mice against PRV-XJ and SVA infection. (A) The survival curves of mice challenged with the PRV-XJ in rPRV-XJ-ΔTK/gE/gI, rPRV-XJ-ΔTK/gE/gI-VP2, and DMEM groups. (B) Pathological observation of H&E staining in brain tissue of mice in each group. The arrows from the top to

virus vector that provides a location for the insertion of foreign genes while reducing the pathogenicity of PRV by deleting gE, gI, and TK genes (34).

As a capsid protein of SVA, the VP2 gene contains conserved sequences. As of 2022, five amino acid sequences of SVA have been published, with only one sequence showing mutation and a low mutation rate (28). Monoclonal antibodies produced by mice immunized with SVA can react with VP2, indicating that VP2 contains potential neutralizing antigenic epitopes (35). In this study, we constructed a recombinant virus, namely, rPRV-XJ-ΔTK/gE/gI-VP2, that expresses the SVA VP2 protein. The appearance of the rPRV-XJ-ΔTK/gE/gI-VP2 virion was not different from that of PRV-XJ. Although the plaque generation capacity of rPRV-XJ-ΔTK/gE/gI-VP2 was lower than that of PRV-XJ, it was comparable to that of rPRV-XJ-ΔTK/gE/gI, which might be due to the deletion of the TK, gE, and gI genes. With the amplification of rPRV-XJ-ΔTK/gE/gI-VP2, the foreign protein was expressed persistently, correctly folded in the host, and exhibited biological activity, demonstrating that gene deletion-attenuated PRV is a suitable vector for expressing VP2.

Mice were selected as animal models to evaluate the safety and immunogenicity of the rPRV-XJ-ΔTK/gE/gI-VP2 vaccine. The deletion of the gE, gI, and TK genes in the recombinant PRV expressing VP2 was found to be safe for mice, which is similar to the ΔTK/gE/gI-PRV strain with three gene deletions (34) or the ΔTK/gE/gI-PRV expressing other foreign proteins (18–22). None of the mice in any titer group showed any apparent clinical symptoms, showed histopathological changes of PRV, or died, and their serum levels of IL-6 and TNF-$\alpha$ did not show a significant increase in this study. The virulent strain PRV-XJ infection in mice can cause nervous signs and systemic inflammation with high levels of IL-6 and TNF-$\alpha$, which are the main causes of death.

Specific CD4 T-cell responses to VP2 were higher than those of VP1 and VP3, indicating that VP2 contains immunodominant T-cell epitopes of SVA (14). In this study, specific T-cell immune responses were activated after rPRV-XJ-ΔTK/gE/gI-VP2 vaccination. The percentage of CD3$^+$ T cells and CD3$^+$ CD4$^+$ T cells increased significantly and was slightly higher than that in rPRV-XJ-ΔTK/gE/gI groups. However, there was no significant difference in the percentage of CD3$^+$ CD8$^+$ T cells compared with that of the rPRV-XJ-ΔTK/gE/gI group. These results suggest that VP2 may have a greater impact on the differentiation of T helper (Th) cells than on cytotoxic T cells. Th cells can be divided into various subtypes, including Th1, Th2, Th17, and others. Th1 cells produce IFN-$\gamma$, IL-2 and TNF-$\alpha$, evoking cell-mediated immunity and phagocyte-dependent inflammation. Th2 cells produce IL-4, IL-5, and IL-13, evoking strong antibody responses (36). To further analyze the subtypes and activity of CD3$^+$ CD4$^+$ Th cells, the IL-2 and IL-4 expression levels were tested by ELISA. In this study, the levels of IL-2 and IFN-$\gamma$ were similar in both vaccine groups, while the IL-4 level in the rPRV-XJ-ΔTK/gE/gI-VP2 group was slightly higher than that in the rPRV-XJ-ΔTK/gE/gI group. It has been reported that the VP2 protein of bluetongue virus (BTV) contains potential IL-4-inducing amino acid peptide segments; however, this induction effect of SVV VP2 needs further investigation (37). In addition, in the rPRV-XJ-ΔTK/gE/gI-VP2 group, inactivated SVA stimulated the proliferation of splenic lymphocytes, but this effect was not observed in the rPRV-XJ-ΔTK/gE/gI group. These results suggest that VP2 may promote the production of specific recognition receptors for SVA on lymphocytes.

A study has demonstrated that mice can serve as a suitable animal model for an initial evaluation of the immunogenicity and protective efficacy of vaccines against SVA (38). Oral challenge with $2 \times 10^{-7}$ TCID$_{50}$ of SVA CH-HNCY-2019 can cause significant pathological changes in the heart, lung, liver, spleen, kidney, brain and duodenum tissue of mice. In this study, mice were inoculated with 50 $\mu$L of 10$^5$ TCID$_{50}$ SVA via nose to simulate

**FIG 6 Legend (Continued)**
the bottom show neuron phagocytosis, vacuolate neuronal degeneration, and nuclear cleavage. (C) The serum IL-6 and TNF-$\alpha$ levels were detected by ELISA in each group. (D) Gene copies of SVA 3D in various tissue of 50 $\mu$L 10$^5$ TCID$_{50}$ SVA intranasal infected-mice. (E) Comparison of SVA 3D gene copies in heart, liver, spleen, and lung tissue of mice in the rPRV-XJ-ΔTK/gE/gI-VP2 and DMEM groups. (F) Pathological observation of H&E staining in the heart and liver tissue of mice in the rPRV-XJ-ΔTK/gE/gI-VP2 and DMEM groups. The arrows from the top to the bottom show endocardial hemorrhage, inflammatory cell infiltration, myocardial cell edema in heart tissue and hepatic vein congestion and hemangiectasis, inflammatory cell infiltration, and hepatocyte balloon-like degeneration of mice in the DMEM group. Arrows from rPRV-XJ-ΔTK/gE/gI-VP2 show diapedesis in heart tissue and loose cytoplasm in liver tissue. Data are presented as mean ± SD ($n = 3$); *, $P < 0.05$; **, $P < 0.01$.

SVA oronasal transmission in pigs. After SVA infection, the mice did not show obvious clinical symptoms or die, but qPCR assays showed that they had transient SVA infection, with gene copies peaking at 3 to 6 days postchallenge and falling below the sensitivity threshold at 14 days postchallenge (dpc). Gene copies in the heart, liver, spleen, and lung tissues were more regular and had a higher level, consistent with the tropism of SVA for various organs in naturally infected newborn piglets (39). Histopathological examination showed that SVA can cause cardiac hemorrhage, endocarditis, balloon-like degeneration, and cytoplasmic vacuolation of hepatocytes, similar to SVA CH-HNCY-2019-infected mice (38). However, no significant pathological changes were observed in other tissues of SVA-infected mice, which may be due to differences in virus strain, challenge dose, and sample collection time. The lack of multiple SVA strains for a challenge comparison or different titer challenge dose comparison is a limitation of this study. In this study, vaccinating mice with rPRV-XJ-ΔTK/gE/gI-VP2 shortened the SVA shedding period and alleviated pathological injury, proving that this recombinant PRV expressing SVA VP2 can protect mice from histopathological damage and viral replication of SVA. SVA detection was not performed in mice immunized with rPRV-XJ-ΔTK/gE/gI because there was no significant effect of inactivated SVA stimulation on splenic lymphocytes in this group. Moreover, similar to rPRV-XJ-ΔTK/gE/gI, rPRV-XJ-ΔTK/gE/gI-VP2 can protect mice from virulent strain PRV-XJ invasion. Injection of PRV-XJ caused evident clinical symptoms in nonimmune mice, while vaccinated mice remained asymptomatic and survived throughout the observation period.

In conclusion, we have developed a live carrier vaccine, namely, rPRV-XJ-ΔTK/gE/gI-VP2, which has shown promising results in model mice. This vaccine is able to induce high levels of neutralizing antibodies against both PRV and SVA, providing complete protection against virulent PRV strains and reducing SVA virus copies while mitigating the pathological damage caused by SVA infection. Furthermore, assessing the safety and immunogenicity of this recombinant strain in the natural host mouse model of PRV and SVA can serve as a reference for vaccine evaluation studies in pigs.

## MATERIALS AND METHODS

**Cells, viruses, and plasmids.** BHK-21, PK-15, and IBRS-2 cells were cultured in DMEM (Gibco; 12100046) supplemented with 10% fetal bovine serum (FBS) (Gibco; 10099133C) at 37°C in a 5% $CO_2$ cell incubator. The PRV-XJ strain (GenBank MW893682.1), rPRV-XJ-ΔgE/gI/TK strain, and SVA strain (GenBank MH716015.1) were preserved at our lab (the Animal Biotechnology Center at the College of Veterinary Medicine, Sichuan Agricultural University). The single guide RNA (sgRNA) targeting TK gene (CTCGACGGCGCCTACGGCAC), sgRNA targeting gE gene (GGGCAGGAACGTCCAGATCC), and the plasmid named pEGFP-gI/28K containing gI and 28K homologous arms were constructed by our lab.

**Construction of recombinant transfer plasmids.** The SVA VP2 gene (852 bp, encoding 284 amino acids [aa]) was amplified by specific primers (Sangon, Shanghai) and inserted into the Xho-I and Sal-I restriction sites of the pEGFP-gI28K vector. The eukaryotic transfer vector pEGFP-gI28K-SVA-VP2 was constructed using the seamless cloning kit (Beyotime, Shanghai) and was extracted using the end-free plasmid minikit II (Omega). The accuracy of the plasmid was verified by enzyme digestion and sequencing.

**Construction of the recombinant virus.** The pEGFP-gI28K-SVA-VP2 plasmid (2.5 $\mu$g) and sgRNA lentiCRISPR V2 plasmid targeting gE (2.5 $\mu$g) were cotransfected into BHK-21 cells at 70% confluence in 6-well plates using the Lipofectamine 3000 transfection reagent (L3000015; Thermo) (7.5 $\mu$L). After 24 h, the BHK-21 cells were infected with 10 $\mu$L of PRV-XJ-ΔTK and cultured at 37°C until 80% cytopathic effect (CPE) was observed. The cell supernatant was harvested to infect BHK-21 cells at 80% confluence. Three rounds of limited dilution in 96-well plates and virus plaque purification in 6-well plates were performed to obtain rPRV-XJ-ΔTK/gE/gI-VP2, with EGFP green fluorescence as a screening condition.

**PCR identification of rPRV-XJ-ΔTK/gE/gI-VP2.** PCR identification of rPRV-XJ-ΔTK/gE/gI-VP2 was carried out using differential primers to verify the lack of gE, gI, and TK and the insertion of SVA VP2. PRV-XJ and double-distilled water (ddH2O) were used as positive and negative controls, respectively. The following primers were used for the PCR: gE forward, 5′-ATCTGGACGTTCCTGCCC-3′; gE reverse, 5′-GTAGATGCAGGGCTCGTACA-3′; gI forward, 5′-TCGCCGAGCAACTACAGCGG-3′; gI reverse, 5′-CGGCGTCGTCGT CTCCGCGT-3′; TK forward, 5′-GATGACATACACATGGCTTTATACGCGCC-3′; TK reverse, 5′-TCACCGCCGCGGCC CGGCGACGTACTC-3′; VP2 forward, 5′-GATGGATCCGATCACAATACCGAAGAA-3′; and VP2 reverse, 5′-CGCTCG AGCTGTTCCTCGTCCGTCC-3′.

To verify the presence of PRV, gB primers were used for amplification, with the specific primer sequences as follows: 5′-GGCTACAGGGCGTCGGGGTC-3′ and 5′-TTCCGCCCGCTGATCTCGAAC-3′.

**Western blotting.** To verify the expression of VP2, gB, and $\beta$-actin, rPRV-XJ-ΔTK/gE/gI-VP2 or PRV-XJ-infected BHK-21 cells were cultured in DMEM with 2% FBS for 36 h. The cells were harvested using ice-cold radioimmunoprecipitation assay (RIPA) buffer containing 1 mM phenylmethylsulfonyl fluoride (PMSF) (Beyotime), and the supernatant was collected for protein extraction. The protein samples were separated by 12% SDS-PAGE

and transferred onto a polyvinylidene difluoride (PVDF) membrane. The membrane was blocked with 5% skim milk at 37°C for 1 h and then incubated with primary antibodies, including a 1:200 dilution of mouse anti-VP2 polyclonal antibody (our lab), a 1:200 dilution of mouse anti-gB polyclonal antibody (our lab), and a 1:3,000 dilution of mouse anti-$\beta$-actin monoclonal antibody (66009-1-Ig; Proteintech) at 4°C for 16 h. The membrane was subsequently incubated with a 1:5,000 dilution of horseradish peroxidase (HRP)-conjugated goat anti-mouse monoclonal antibody (Bio-Rad; 5178-2504) at 37°C for 1 h. Finally, the membrane was treated with Clarity Western ECL substrate (Bio-Rad; 170-5061) and imaged using a chemiluminescence imaging system.

**Immunofluorescence assay (IFA).** A total of 10 $\mu$L of rPRV-XJ-ΔTK/gE/gI-VP2 or PRV-XJ infected BHK-21 cells were cultured in DMEM containing 2% FBS in 6-well plates for 24 h. The cells were then fixed with immobile liquid (methanol and acetone, 1:1) at 4°C for 30 min, followed by blocking with 5% bovine serum albumin (BSA) at 20°C for 1 h. The cells were then incubated with primary antibody (1:200 dilution of mouse anti-VP2 polyclonal antibody) at 4°C for 16 h and with secondary antibody (1:500 dilution of Corallite 594-conjugated goat anti-mouse IgG [SA00013-3; Proteintech]) at 37°C for 1 h. 4′,6-Diamidino-2-phenylindole (DAPI; C0060; Solarbio) was used for nuclear staining.

**Replication kinetics of rPRV-XJ-ΔTK/gE/gI-VP2.** A 10-fold serial diluent (0.025 mL) of PRV-XJ, rPRV-XJ-ΔTK/gE/gI, or rPRV-XJ-ΔTK/gE/gI-VP2 was taken to inoculate 80% confluent BHK-21 cells in 96-well plates. Then the cells were cultured at 37°C until CPE occurred. The virus titer was measured using the Reed-Muench method, with rPRV-XJ $TCID_{50}$ of $10^{-7.98}$/0.025 mL, rPRV-XJ-ΔTK/gE/gI $TCID_{50}$ of $10^{-7.28}$/0.025 mL, and rPRV-XJ-ΔTK/gE/gI-VP2 $TCID_{50}$ of $10^{-7.13}$/0.025 mL. For the virus one-step growth curve, 0.1 multiplicity of infection (MOI) of PRV-XJ, rPRV-XJ-ΔTK/gE/gI, or rPRV-XJ-ΔTK/gE/gI-VP2 was used to inoculate BHK-21 cells in 12-well plates. The cells and supernatant were harvested at 2, 4, 6, 8, 10, 12, 24, 36, 48, 60, and 72 h postinfection, with three samples taken at each time point as replicates. The virus one-step growth curve was calculated using the Reed-Muench method. To evaluate genetic stability, rPRV-XJ-ΔTK/gE/gI-VP2 was passaged 21 times in BHK-21 cells; the lack of gE, gI, and TK and the insertion of VP2 were verified by PCR in the filial generations F5, F10, F15, and F21.

**Plaque assays.** A total of 100 $TCID_{50}$ PRV-XJ, rPRV-XJ-ΔTK/gE/gI, or rPRV-XJ-ΔTK/gE/gI-VP2 was taken to inoculate 80% confluent BHK-21 cells in 6-well plates, followed by incubation at 37°C until CPE occurred. The cells were then fixed and stained with formalin-crystal violet fixed staining solution, and the number and area of plaques were determined automatically using IPP6.0 software.

**Transmission electron microscopic observation of virions.** BHK-21 cells inoculated with PRV-XJ or rPRV-XJ-ΔTK/gE/gI-VP2 were collected and fixed with a 3% glutaraldehyde fixative. After fixation, the cell mass was processed by Chengdu Lilai Biotechnology Co., Ltd., and transmission electron microscopy images of virions were captured.

**Safety and immunogenicity assessment in mice.** Six-week-old female BALB/c mice were divided randomly into five groups ($n = 10$/group). The positive-control group was inoculated with 200 $\mu$L of $10^3$ $TCID_{50}$ of PRV-XJ, while the negative-control group was inoculated with DMEM instead. Three test groups were inoculated with $10^5$ $TCID_{50}$, $10^6$ $TCID_{50}$, and $10^7$ $TCID_{50}$ of rPRV-XJ-ΔTK/gE/gI-VP2. After continuous observation for 14 days, the survival curves of the mice were plotted. At 3 days postvaccination, tail venous blood was collected to detect the levels of IL-6 and TNF-$\alpha$, and the brains were collected for H&E staining.

**Shedding of rPRV-XJ-ΔTK/gE/gI-VP2 in mice.** Mouse feces were collected every 2 days following vaccination. These samples were analyzed to determine the shedding of recombinant viruses in mice. The shedding of the vaccine virus in the mouse model was evaluated using qPCR primers specific to PRV IE180 (PRV IE180 forward, TAAGTCCGGCTACAGCACCAAGTCC; and reverse, TCTTCGTCGTCGCGGTGGCCGT).

**Histopathology assay.** The tissue was fixed with 4% paraformaldehyde for 96 h, embedded with paraffin, and then cut into 4-mm sections. The paraffin sections were stained with hematoxylin and eosin and examined under a microscope.

**Mouse immunization and challenge experiment.** The immunization/virus challenge experiments are represented graphically in Fig. 4A. Six-week-old female BALB/c mice were divided randomly into three groups. The negative-control group was inoculated intranasally with DMEM, while the test group ($n = 43$/group) was inoculated with $10^6$ $TCID_{50}$ rPRV-XJ-ΔTK/gE/gI-VP2 and the positive-control group ($n = 19$/group) was inoculated with $10^6$ $TCID_{50}$ rPRV-XJ-ΔTK/gE/gI for the PRV challenge experiment. Booster immunization was performed at 14 days postvaccination (dpv). Spleen samples ($n = 3$/group) were collected from each group for flow cytometry analysis and proliferation tests, and blood samples were collected weekly until 49 dpv. At 28 dpv, mice of each group were inoculated with 200 $\mu$L $10^5$ $TCID_{50}$ ($n = 10$/group) PRV-XJ via footpad or 50 $\mu$L $10^5$ $TCID_{50}$ SVA via nose ($n = 18$/group).

The survival curve of mice was plotted, and after 3 days post-PRV challenge (dpc), blood samples were collected from the mice for IL-6 and TNF-$\alpha$ analysis. The mice were observed for 14 days, and brain tissue was collected for histopathological examination. Additionally, heart, liver, spleen, lung, kidney, brain, intestine, and mesenteric lymph node samples of mice inoculated with SVA in each group were collected on days 1, 3, 6, 10, 14, and 21 for histopathology examination and SVA viral load tests. Replicates were taken from 3 mice at each time point. All experimental procedures were reviewed and approved by the Sichuan Agriculture University Animal Care and Use Committee (license number SCXK [Sichuan] 2013-0001).

**ELISA for PRV-gB-specific antibody, SVA-VP2-specific antibody, and cytokines.** Serum was extracted from the blood samples (50 $\mu$L) for ELISA analysis. The PRV-gB-specific antibody level was evaluated by the ID.vet commercial ELISA kit (France). The levels of mouse IL-2, IL-4, IFN-$\gamma$, TNF-$\alpha$, and IL-6 were evaluated by a Neobioscience commercial ELISA kit (China). For the evaluation of SVA-specific VP2 antibodies, an ELISA method was established. The SVA-VP2 gene fragment was inserted into PET32a (Novagen) and transformed into the BL21(DE3) *Escherichia coli* strain. The recombinant protein was induced with 0.5 mmol/L isopropyl-$\beta$-D-thiogalactopyranoside (IPTG; Solarbio), and the strain was cultured for 48 h at 16°C and 100 rpm. The optimal antigen coating concentration was 62.5 ng/mL, and the serum was diluted 1:200. After incubation for 1.5 h at 37°C, a secondary antibody (1:5,000 diluted HRP-conjugated goat anti-mouse IgG) was added and

incubated for 1 h at 37℃. The TMB substrate chromodeveloping solution (Solarbio) was colored for 15 min, followed by the addition of 2 M $H_2SO_4$ to each well to terminate the reaction. The absorbance value was detected at an opical density at 450 nm ($OD_{450}$) using a microplate analyzer (Bio-Rad) within 15 min.

**Neutralizing antibody assay in mice.** From the blood samples, 100 $\mu$L serum was extracted and inactivated by neutralizing antibody assay. The serum was then serially diluted 2-fold and mixed with an equal volume of 100 PFU of either SVA or PRV-XJ at 37℃ for 1 h. After the neutralization reaction, the viral-serum mixture was used to inoculate 80% confluent BHK-21 cells in 6-well plate, which were then cultured at 37℃ until the appearance of CPE. Negative serum from mice was used as a control. The 6-well plate was fixed and stained with formalin-crystal violet staining solution, and the number and area of plaques were automatically read by IPP6.0 software. The serum neutralization titer was measured by plaque reduction assay, and the serum dilution that reduced the number of plaques by 50% was used as the neutralization titer for the sample. Proportion was used as the calculation method for the plaque reduction assay. For instance, if the percentage of plaque reduction of the serum sample under the serum dilution at $2^4$ is 53.85%, the percentage of plaque reduction at the serum dilution at $2^5$ is 30.77%, then the neutralizing antibody titer can be calculated as follows: neutralizing antibody titer = $(53.85 - 50)/(53.85 - 30.77) \wedge (25 - 24) + 2^4 = 2^{4.22}$.

**Flow cytometry analysis of CD3/CD4/CD8 murine splenic lymphocytes.** Splenic cells were obtained from the spleen samples by passing them through 100-mesh cell filtration sieves. The cells were then lysed with ammonium-chloride-potassium (ACK) lysis buffer at 4℃ for 5 min. Next, $1 \times 10^6$ splenic lymphocytes in phosphate-buffered saline (PBS) were prepared and stained with FITC anti-mouse CD3 antibody, APC anti-mouse CD4 antibody, and PE anti-mouse CD8a antibody (100204, 100412, and 100708, respectively; Biolegend) at a 1:250 dilution for 30 min in the dark at 4℃. The stained samples were analyzed using a BD FACSVerse flow cytometer, and the data were collected and analyzed using Flowjo 10 software.

**Murine splenic lymphocyte proliferation test.** The splenic cells were diluted to $5 \times 10^6$ cells/mL with 10% FBS + 1640 medium in 96-well plates at 100 $\mu$L/well. The cells were then treated with 10 $\mu$g/mL of ConA as a positive control, 1640 as a negative control, UV-inactivated $10^6$ $TCID_{50}$/mL SVA as test group, and 10% FBS + 1640 medium without cells as the blank control. The proliferation of spienic lymphocytes was evaulated using CCK-8 kit (Beyotime) and the absorbance was detected at $OD_{450}$. The stimulation index (SI) of ConA or inactivated SVA = (postive control or test group $OD_{450}$ - blank control $OD_{450}$)/(negative control $OD_{450}$ - blank control $OD_{450}$).

**SVA viral load assay in mice.** Tissue samples weighing 100 mg were ground with DMEM and lysed using the TRIzol kit (TaKaRa) for RNA extraction. The RNA sample was dissolved with 30 $\mu$L diethyl pyrocarbonate (DEPC) water, and the concentration was measured by using a nucleic acid concentration analyzer (Scandrop200; Analytik Jena, Germany). Reverse transcription was performed using the Prime Script II 1st strand cDNA synthesis kit (TaKaRa), and RT-qPCR was performed to determine the transcription of 3D using the Lightcycler 96 instrument (Roche) and TB green premix *Ex Taq* II (Tli RNaseH Plus) (TaKaRa). The specific primers for 3D amplification were designed by Oligo 7.0 and are as follows: 3D forward, 5'-GAGAAAGACGACCGCACAC-3'; and reverse, 5'-AAGGAGAAAACCCGATGAG -3'.

**Statistical analysis.** Statistical analysis was performed using Prism 8.0 (GraphPad Software, USA) to determine significant differences in the data. Paired Student's *t* test and one-way analysis of variance (ANOVA) were used to test for differences between different groups. *P* values less than 0.05 were considered significant, and *P* values less than 0.01 were considered highly significant. Histograms were generated to visualize the data.

**Data availability.** The original contributions presented in the study are included in the article, and further inquiries can be directed to the corresponding author.

## ACKNOWLEDGMENTS

This work was supported by the Chongqing Municipal Technology Innovation and Application Development Project (no. cstc2021jscx-dxwt BX0007), the Key K&D Program of Sichuan Science and Technology Plan (no. 2022YFN0007), the Porcine Major Science and Technology Project of Sichuan Science and Technology Plan (no. 2021ZDZX0010-3), the Sichuan Provincial Department of Science and Technology Rural Area Key R&D Program (no. 2020YFN0147), and the Agricultural Industry Technology System of Sichuan Provincial Department of Agriculture (no. CARS-SVDIP).

Q.T., Z.X., and L.Z. contributed to conceptualization. Q.T., L.X., Y.Y., Y. Zhang, Z.L. contributed to formal analysis. Q.T. contributed to data curation and original draft preparation. Q.T., L.X., and T.X. contributed to software application. J.W. and L.D. contributed to supervision. Y. Zhou contributed to resources. Z.X. contributed to validation and resources. L.Z. contributed to project administration and funding acquisition. All authors have read and agreed to the published version of the manuscript.

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
