## [Reviewer comments · Microbiology Spectrum]

Microbiology Spectrum

The construction and immunogenicity analyses of a recombinant pseudorabies virus with Senecavirus A VP2 protein co-expression

Qian Tao, Ling Zhu, Lei Xu, Yanting Yang, Yang Zhang, Zheyang Liu, Tong Xu, Jianhua Wen, Lishuang Deng, Yuancheng Zhou, and Zhiwen Xu

Corresponding Author(s): Zhiwen Xu, Sichuan Agricultural University

Review Timeline:

Submission Date:	December 21, 2022
Editorial Decision:	February 7, 2023
Revision Received:	March 2, 2023
Accepted:	March 6, 2023

Editor: Leonidas Stamatatos

Reviewer(s): The reviewers have opted to remain anonymous.

Transaction Report:

DOI: <https://doi.org/10.1128/spectrum.05229-22>

February 7, 2023

Prof. Zhiwen Xu
Sichuan Agricultural University
No. 211 , Huimin Road , 11 Wenjiang District, Chengdu City, Sichuan Province , C
Chengdu, Please select
China

Re: Spectrum05229-22 (**The construction and immunogenicity analyses of a recombinant pseudorabies virus with Senecavirus A VP2 protein co-expression**)

Dear Prof. Zhiwen Xu:

Link Not Available

Sincerely,

Leonidas Stamatatos

Journals Department
Reviewer comments:

Reviewer #1 (Comments for the Author):

The paper by Tao et al describes the development of a live attenuated recombinant pseudorabies virus vaccine that expresses the VP2 gene of Senecavirus. They characterize the virus in vitro and use it to immunize mice against pseudorabies and Senecavirus infection. They characterize the immune response to vaccination examine viral load and histopathology in animals. In the process, they describe the effect of Senecavirus in mice. The experimental methods are technically sound and the data are logical. However there are some issues/questions.

1. The manuscript would benefit from proof-reading. There are grammatical errors and typos throughout that affect read-ability.

In addition, many abbreviations are used without ever being defined or if they are defined, the reader needs to use the search function. Examples include but are not limited to lines 63, 149, and 274.

2. Abstract line 16 states that SVA VP2 activates a Th2 response to enhance the humoral response to SVA. I am not sure there is enough data to fully support this statement since Th1 cytokines like IFN gamma and IL2 are also produced. In the discussion, line 263, this conclusion is also made with the statement that IL-4 levels in the vaccine group was higher than the rPRV-XJ-deltaTK/gE/gI group. However, this was not a statistically significant difference. This is also a confusing sentence, because what does "vaccine group" refer to? rPRV-XJ-deltaTK/gE/gI is also a vaccine group, but I assume "vaccine group" refers to the vaccine expressing VP2.

3. In the animal experiments, is the vaccine virus shed and if so for how long?

4. Do the mice infected with Senecavirus exhibit any clinical symptoms from cardiac disease? What are the main clinical manifestations of Senecavirus in pigs and are they similar to mice? Further discussion of this could be helpful to see how well the mouse model recapitulates disease in the target population.

Minor

1. Figure 4 does not clearly state the route of immunization and challenge. Searching the methods suggest intranasal immunization but stating this clearly in figure legend or in the section referring to the figure would be helpful.

2. The methods section, line 411, mention a "proportion method". What is this method? Also, the y-axis of Figure 4 shows neutralization titer in log base 2, but it is not clear what the numbers refer to? Is this the dilution of serum that leads to 50% plaque reduction?

Reviewer #2 (Comments for the Author):

Article by Tao et al.,
Microbiology Spectrum

Presented manuscript is devoted to the development of live attenuated virus vaccine against Senecavirus A (SVA) and pseudorabies virus (PRV), highly contagious swine diseases that present a substantial burden for agricultural industry. Despite its importance, presented study has several issues outlined below.

Major points:

1. Figure 2D: There is about one log difference in virus titer between parental strain and the recombinant virus at 0h post-infection. How is that possible if same infectious dose of 0.1 MOI was used to infect cells? How was virus infectivity assessed? It is not described in Methods! Was cell density different? The difference in virus titer remains stable throughout the experiment and at 48h post infection the titers of both viruses become almost identical. This suggests that the difference observed in titer is due to different starting dose rather than a result of deletion of 3 genes. I cannot agree with statement that kinetic of virus growth is different. Growth kinetics is reflected by curve slope and shape which apparently very similar for both viruses. However, formal analysis of curve slope and parallelism can provide a sound answer to this question.

Methods section indicates that three viruses were used for growth kinetic experiment: PRV-XJ, rPRV-XJ-deltaTK and rPRV-deltaTK/gE/gI-VP2. rPRV-XJ-deltaTK is not shown on Figure 2D. What is the reason for not including this virus into the figure? There is no description how virus titer was measured. Methods only says that cells and supernatant were harvested.

2. How exactly size of plaques was measured? The technique is not described! Could different plaque size be a result of the virus dose?

3. Transmission electron microscopy is not described in Methods.

4. Histopathology technique is not described in Methods.

5. Is mice survival the only measure of safety of recombinant virus created? Apparently, deletion of gE, gI and TK affects virus virulence and transmissibility. As such, appropriate references should be provided and subsequently discussed.

6. The following statement requires a reference: "IFN-g can be used as a standard to evaluate the cell-mediated immune effects of vaccines; it plays an important role in innate and adaptive immunity".

7. Line 143-144: "the level of IFN-g improved significantly..." Was it reduced/diminished in the first place to be improved after vaccination? Did authors meant "increased"?

8. Authors need to provide a reason why ConA-induced stimulation was used as a comparison?

9. "IL-2 was secreted by Th1 cell, IL-4 was secreted by Th2 cell." This statement implies that authors investigated Th1 and Th2 cells defined by flow cytometry or other methods. It seems that this phrase is introductory to the following experiment where IL-2 and IL-4 serum levels were investigated. If this is the case, the sentence to be revised and supplemented with appropriate reference.

10. Lines 173-174: Complete lack of flow. It is absolutely unclear why data on SVA in mouse tissues are presented here? In the preceding paragraph, only immunization and infection with PRV is described! Where these mice infected with SVV also or it was a separate experiment? Experiment testing SVA infectivity and consequences of infection is not described raising concerns about study integrity. Moreover, there is a statement that it is not known if SVV can infect mice. However, simple search yields a reference doi:10.1016/j.rvsc.2021.12.010 providing evidence that SVV in fact infects mice.

11. What is 3D RNA is not explain either.

12. Graphical representation of immunization/virus challenge experiment would be helpful to understand study design. Were both viruses given simultaneously, how much time was between immunization and virus challenge?

13. Discussion is poorly written, extensively long and not focused. Some if it, the first two paragraphs, for example, is a repeat of Introduction. Several following paragraphs is a simple repeat of the results observed.

In lines 262-263 authors state: "while the IL-4 level in the recombinant vaccine group was higher than rPRV-XJ- Δ TK/gE/gI group". According to Figure 5D this difference was not statistically significant and authors admit that in the corresponding statement on lines 156-158. Therefore, the following implication of "improved" IL-4 levels via activation of TH2 cells and the following enhancement of humoral response seems speculative and is not supported by data provided. In contrast, presented results rather show no difference between IL-2 and IL-4 that is well in line with general understanding of immune response to viral infection that in fact immunization with recombinant virus is.

Line 269: As mentioned before, there is published study about SVA infection in mice. This should be discussed in regards with data presented in this manuscript.

Study limitations are not discussed.

14. Can the recombinant virus created in this study be transmitted between animals?

Minor points:

1. Editorial help with English language is required.

2. Where animal experiments reviewed by Institutional Review Board or other authorized entity for humane animal use?

3. Line 84: Explanation of what are F5, F10, F15, F21 is needed.

4. Figure 2B: what is Mock? Explanation in figure legend is needed.

5. Figure 2B: while beta-actin provides a control for cell proteins, another control for viral protein(s) would be helpful here.

6. Fig. 4A and 4B: What is the reason of showing results of anti-gB ELISA as signal-to-noise ratio and VP2 ELISA in absorbance units?

7. Fig.5A is not informative and rather shows raw data. It can be moved into Supplementary materials.

8. Fig. 6A, 6B and 6C replicate data provided in Fig.3. These data can be moved into Supplementary materials with reference in text that survival kinetic and brain tissue damage as well as serum IL-6 and TNF levels were similar to that found in safety experiment.

Staff Comments:

Preparing Revision Guidelines

- Point-by-point responses to the issues raised by the reviewers in a file named "Response to Reviewers," NOT IN YOUR COVER LETTER.

- Upload a compare copy of the manuscript (without figures) as a "Marked-Up Manuscript" file.

- Each figure must be uploaded as a separate file, and any multipanel figures must be assembled into one file.

- Manuscript: A .DOC version of the revised manuscript

- Figures: Editable, high-resolution, individual figure files are required at revision, TIFF or EPS files are preferred

Please return the manuscript within 60 days; if you cannot complete the modification within this time period, please contact me. If you do not wish to modify the manuscript and prefer to submit it to another journal, please notify me of your decision immediately so that the manuscript may be formally withdrawn from consideration by Microbiology Spectrum.

Manuscript Number (Spectrum05229-22): The construction and immunogenicity analyses of a recombinant pseudorabies virus with Senecavirus A VP2 protein co-expression

Dear Editor and Reviewers,

Thank you for giving us the opportunity to resubmit our revised manuscript. We appreciate the concerns and suggestions provided by the reviewers and editor, and we have revised our manuscript accordingly. The modified parts in the text were indicated with red color, and the responses to the reviewers are as follows. These suggestions helped us improve the manuscript, and we hope that you find it suitable for publication.

I look forward to hearing from you soon.

Yours sincerely,

Zhi-Wen Xu

Reviewers' comments:

Reviewer #1:

Major concerns:

Comment 1: The manuscript would benefit from proof-reading. There are grammatical errors and typos throughout that affect read-ability. In addition, many abbreviations are used without ever being defined or if they are defined, the reader needs to use the search function. Examples include but are not limited to lines 63, 149, and 274.

Response: Thanks for your question. The grammatical errors and typos have been professionally corrected, and the abbreviations have been defined in this article.

Comment 2: Abstract line 16 states that SVA VP2 activates a Th2 response to enhance the humoral response to SVA. I am not sure there is enough data to fully support this statement since Th1 cytokines like IFN gamma and IL2 are also produced. In the discussion, line 263, this conclusion is also made with the statement that IL-4 levels in the vaccine group was higher than the rPRV-XJ-deltaTK/gE/gI group. However, this was not a statistically significant difference. This is also a confusing sentence, because what does "vaccine group" refer to? rPRV-XJ-deltaTK/gE/gI is also a vaccine group, but I assume "vaccine group" refers to the vaccine expressing VP2.

Response: Thanks for your question. I apologize for my imprecise inference in the abstract, and I have removed it. In the discussion section, the inference has been revised and marked with red color. The modified content is as follows: It has been reported that VP2 protein of Bluetongue Virus (BTV) contains potential

IL-4-inducing amino acid peptide segments, however, this induction effect of SVV VP2 needs further investigation(1) (line 269-272). Additionally, I have revised the term "recombinant vaccine group" in line 272 to "rPRV-XJ-ΔTK/gE/gI-VP2 group."

Comment 3: In the animal experiments, is the vaccine virus shed and if so for how long?

Response: Thanks for your question. The shedding of the vaccine virus in mice model was evaluated using qPCR. Following single footpad inoculation with 10^7 TCID₅₀ of rPRV-XJ-ΔTK/gE/gI-VP2, the PRV IE180 gene copies peaked at 7 days post-vaccination (dpv), and fell below the sensitivity threshold by 15 dpv. These modified contents have been marked with red color in **RESULTS** (line 111-114) and **MATERIALS AND METHODS** (line 407-412).

Comment 4: Do the mice infected with Senecavirus exhibit any clinical symptoms from cardiac disease? What are the main clinical manifestations of Senecavirus in pigs and are they similar to mice? Further discussion of this could be helpful to see how well the mouse model recapitulates disease in the target population.

Response: Thanks for your question. The mice infected with Senecavirus survived without obvious clinical symptoms from cardiac disease throughout the experiment. The clinical characteristics of SVA infection mainly include lethargy, anorexia, acute lameness, and vesicular lesions on the snouts, coronary bands and hoof in pigs. A study has demonstrated that mice can serve as a suitable animal model for initial evaluation of the immunogenicity and protective efficacy of vaccines against SVA. Orally challenged with 2×10^{-7} TCID₅₀ of SVA CH-HNCY-2019 can cause

significant pathological changes in the heart, lung, liver, spleen, kidney, brain and duodenum tissue of mice(2)(line 277-281). Gene copies in the heart, liver, spleen, and lung tissues were more regular and had a higher level, consistent with the tropism of SVA for various organs in naturally infected newborn piglets(3)(line285-288). This result also provides support for mice to be used as animal models of SVA infection. These modified contents have been marked with red color in **DISCUSSION**.

Minor concerns:

1. Figure 4 does not clearly state the route of immunization and challenge. Searching the methods suggest intranasal immunization but stating this clearly in figure legend or in the section referring to the figure would be helpful.

Response: Thanks for your question. The route of immunization and challenge has been stated in figure legend and marked with red color. The modified content is as follows: Figure 4 Immune responses after intranasal immunization with recombinant virus (line 705). Figure 6 (D) Gene copies of SVA 3D in various tissue of $50 \text{ ul } 10^5 \text{ TCID}_{50}$ SVA intranasal infected-mice (line 723-724).

2. The methods section, line 411, mention a "proportion method". What is this method? Also, the y-axis of Figure 4 shows neutralization titer in log base 2, but it is not clear what the numbers refer to? Is this the dilution of serum that leads to 50% plaque reduction?

Response: Thanks for your question. The serum neutralization titer was measured by plaque reduction assay, the serum dilution that reduced the number of plaques by 50% was used as the neutralization titer for this serum sample. Proportion was used as the

calculation method of plaque reduction assay. For example, the percentage of plaque reduction of the serum sample under the serum dilution at 2^4 is 53.85%, the percentage of plaque reduction at the serum dilution at 2^5 is 30.77%, neutralizing antibody titer = $(53.85-50) / (53.85-30.77) \wedge (2^5-2^4) + 2^4 = 2^{4.22}$. These modified contents have been marked with red color in **MATERIALS AND METHODS** (line 459-466).

Reviewer #2:

Major concerns:

Comment 1: Figure 2D: There is about one log difference in virus titer between parental strain and the recombinant virus at 0h post-infection. How is that possible if same infectious dose of 0.1 MOI was used to infect cells? How was virus infectivity assessed? It is not described in Methods! Was cell density different? The difference in virus titer remains stable throughout the experiment and at 48h post infection the titers of both viruses become almost identical. This suggests that the difference observed in titer is due to different starting dose rather than a result of deletion of 3 genes. I cannot agree with statement that kinetic of virus growth is different. Growth kinetics is reflected by curve slope and shape which apparently very similar for both viruses. However, formal analysis of curve slope and parallelism can provide a sound answer to this question. Methods section indicates that three viruses were used for growth kinetic experiment: PRV-XJ, rPRV-XJ-deltaTK and rPRV-deltaTK/gE/gI-VP2. rPRV-XJ-deltaTK is not shown on Figure 2D. What is the reason for not including

this virus into the figure? There is no description how virus titer was measured. Methods only says that cells and supernatant were harvested.

Response: Thanks for your question. Sorry for my carelessness, the replication kinetics of rPRV-XJ- Δ TK/gE/gI-VP2 has been re-tested, and the virus one-step growth curve of rPRV-XJ- Δ TK/gE/gI has been added into Figure 2D. In addition, the plaque assays of rPRV-XJ- Δ TK/gE/gI has been added into Figure 2E. The test methods for virus TCID₅₀: A 10-fold serial diluent (0.025mL) of PRV-XJ, rPRV-XJ- Δ TK/gE/gI or rPRV-XJ- Δ TK/gE/gI-VP2 was taken to inoculate 80% confluent BHK-21 cells in 96-well plates. Then the cells were cultured at 37°C until CPE occurred. The virus titer was measured using Reed–Muench method, with rPRV-XJ TCID₅₀=10^{-7.98}/0.025ml, rPRV-XJ- Δ TK/gE/gI TCID₅₀=10^{-7.28}/0.025ml, and rPRV-XJ- Δ TK/gE/gI-VP2 TCID₅₀ =10^{-7.13}/0.025ml. These modified contents have been marked with red color in **RESULTS** (line100-105), **MATERIALS AND METHODS** (line375-380) and **DISCUSSION** (line 237-240).

Comment 2: How exactly size of plaques was measured? The technique is not described! Could different plaque size be a result of the virus dose?

Response: Thanks for your question. The test methods of plaque assays: 100 TCID₅₀ PRV-XJ, rPRV-XJ- Δ TK/gE/gI or rPRV-XJ- Δ TK/gE/gI-VP2 was taken to inoculate 80% confluence BHK-21 cells in 6-well plates, followed by incubation under 37°C until CPE occurred. The cells were then fixed and stained with formalin-crystal violet fixed staining solution, and the number and area of plaques were automatically determined using IPP6.0 software. The modified content has been marked with red

color in **MATERIALS AND METHODS** (line389-393).

Comment 3: Transmission electron microscopy is not described in Methods.

Response: Thanks for your question. Transmission electron microscopic observation of virion : BHK-21 cells inoculated with PRV-XJ or rPRV-XJ- Δ TK/gE/gI-VP2 were collected and fixed with a 3% glutaraldehyde fixative. After fixation, the cell mass was processed by Chengdu Lilai Biotechnology Co., Ltd., and transmission electron microscopy images of virions were captured. The modified content has been marked with red color in **MATERIALS AND METHODS** (line394-398).

Comment 4: Histopathology technique is not described in Methods.

Response: Thanks for your question. Histopathology technique: The tissue was fixed with 4% paraformaldehyde for 96 h, embedded with paraffin, and then cut into 4 mm sections. The paraffin sections were stained with hematoxylin and eosin and examined under a microscope. The modified content has been marked with red color in **MATERIALS AND METHODS** (line 413-415).

Comment 5: Is mice survival the only measure of safety of recombinant virus created? Apparently, deletion of gE, gI and TK affects virus virulence and transmissibility. As such, appropriate references should be provided and subsequently discussed.

Response: Thanks for your question. In this study, the serum levels of inflammatory cytokines and the brain tissue histopathology assay of the mice vaccinated with rPRV-XJ- Δ TK/gE/gI-VP2 also were used as the measure of safety assessment. Appropriate references have been provided and subsequently discussed. The modified content has been marked with red color in **DISCUSSION** (line 245-251).

Comment 6: The following statement requires a reference: "IFN-g can be used as a standard to evaluate the cell-mediated immune effects of vaccines; it plays an important role in innate and adaptive immunity".

Response: Thanks for your question. This sentence has been modified and marked with red color: Interferon- γ (IFN- γ) can serve as a criterion for evaluating the specific T cell-mediated immune effects of vaccines (4) (line 153-154).

Comment 7: Line 143-144: "the level of IFN-g improved significantly..." Was it reduced/diminished in the first place to be improved after vaccination? Did authors meant "increased"?

Response: Thanks for your question. This sentence has been modified and marked with red color: Compared with DMEM group, the level of IFN- γ increased significantly in these vaccinated group (line 153).

Comment 8. Authors need to provide a reason why ConA-induced stimulation was used as a comparison?

Response: Thanks for your question. Con A is a phytohemagglutinin with potent mitogenic capacity, which can be used as a positive standard for stimulating splenic lymphocyte proliferation (5). This sentence has been added at line 162-163 and marked with red color.

Comment 9: "IL-2 was secreted by Th1 cell, IL-4 was secreted by Th2 cell." This statement implies that authors investigated Th1 and Th2 cells defined by flow cytometry or other methods. It seems that this phrase is introductory to the following experiment where IL-2 and IL-4 serum levels were investigated. If this is the case, the

sentence to be revised and supplemented with appropriate reference.

Response: Thanks for your question. The modified content has been moved into **DISCUSSION** (line 263-265) and marked with red color: Th1 cells produce IFN- γ , IL-2 and TNF- α , evoking cell-mediated immunity and phagocyte-dependent inflammation. Th2 cells produce IL-4, IL-5 and IL-13, evoking strong antibody responses(6). To further analyzed the subtypes and activity of CD3⁺ CD4⁺ Th cell, the IL-2 and IL-4 expression levels were test by ELISA.

Comment 10: Lines 173-174: Complete lack of flow. It is absolutely unclear why data on SVA in mouse tissues are presented here? In the preceding paragraph, only immunization and infection with PRV is described! Where these mice infected with SVV also or it was a separate experiment? Experiment testing SVA infectivity and consequences of infection is not described raising concerns about study integrity. Moreover, there is a statement that it is not known if SVV can infect mice. However, simple search yields a reference doi:10.1016/j.rvsc.2021.12.010 providing -evidence that SVV in fact infects mice.

Response: Thanks for your question. Sorry for my carelessness, this reference has been cited in our study (line 277-281). To simulate oronasal transmission of SVA in pigs, mice in each group were injected with 50 μ L 10^5 TCID₅₀ of SVA via the nose. After SVA infection, mice in each group did not show obvious clinical symptoms or death. The gene copy numbers of SVA in mouse tissues of each group was detected to evaluate the protection of rPRV-XJ- Δ TK/gE/gI-VP2 on SVA-infected mice. (line 186-191).

Comment 11: What is 3D RNA is not explain either.

Response: Thanks for your question. The 3D gene is a conserved gene of SVA (7), and in this study, it was used as the template to establish qPCR standard curve for SVA detection ($y = -3.4653x + 36.194$, $R^2 = 0.9993$). (line 190-192).

Comment 12: Graphical representation of immunization/virus challenge experiment would be helpful to understand study design. Were both viruses given simultaneously, how much time was between immunization and virus challenge?

Response: Thanks for your question. Graphical representation of immunization/virus challenge experiment has been added into figure 4. The modified content has been marked with red color (line 416-417, line 706). Sorry for the incomplete experimental design on mice, we will take your advice, when conducting vaccine evaluation studies on pigs.

Comment 13: Discussion is poorly written, extensively long and not focused. Some if it, the first two paragraphs, for example, is a repeat of Introduction. Several following paragraphs is a simple repeat of the results observed.

In lines 262-263 authors state: "while the IL-4 level in the recombinant vaccine group was higher than rPRV-XJ- Δ TK/gE/gI group". According to Figure 5D this difference was not statistically significant and authors admit that in the corresponding statement on lines 156-158. Therefore, the following implication of "improved" IL-4 levels via activation of Th2 cells and the following enhancement of humoral response seems speculative and is not supported by data provided. In contrast, presented results rather show no difference between IL-2 and IL-4 that is well in line with general

understanding of immune response to viral infection that in fact immunization with recombinant virus is.

Line 269: As mentioned before, there is published study about SVA infection in mice. This should be discussed in regards with data presented in this manuscript.

Study limitations are not discussed.

Response: Thanks for your question. Sorry for my poorly written. The first paragraph of **Discussion** mainly about epidemiology of SVA in China. The second paragraph of **Discussion** mainly about the inactivated vaccine of SVA which have been report and the drawbacks of inactivated vaccine. **Introduction** of this study mainly about epidemiology of SVA in the world, SVA structural protein, introduction of PRV and the live vector vaccine construction by PRV. The grammatical errors and typos have been corrected by professional.

Sorry for my imprecise inference, the inference has been revised and marked with red color. The modified content is as follows: It has been reported that VP2 protein of Bluetongue Virus (BTV) contains potential IL-4-inducing amino acid peptide segments, however, this induction effect of SVV VP2 needs further investigation(1) (line 269-272).

Sorry for my carelessness, this reference has been cited in our study (line 277-281). It has been discussed and compared with the data presented in this study (line 290-293).

This study limitations have been discussed at line 293-294.

Comment 14: Can the recombinant virus created in this study be transmitted between

animals?

Response: Thanks for your question. We apologize for the limitations of our experimental design. This study did not include experiments on the transmission of the recombinant virus between animals. Although we believe that it has the potential to be transmitted from animal to animal, further experimental verification is required to confirm this.

Minor points:

1. Editorial help with English language is required.

Response: Thanks for your question. Sorry for my poorly written, the grammatical errors and typos have been professionally corrected.

2. Where animal experiments reviewed by Institutional Review Board or other authorized entity for humane animal use?

Response: Thanks for your question. The ethics statement has been added into line 433-435 and marked with red color: All experimental procedures were reviewed and approved by the Sichuan Agriculture University Animal Care and Use Committee (license number SCXK (Sichuan) 2013-0001).

Line 84: Explanation of what are F5, F10, F15, F21 is needed.

Response: Thanks for your question. To assess the stability of the recombinant virus, it is necessary to perform continuous passage for more than 20 generations and extract DNA from the F5, F10, F15, and F21 generations for PCR identification. This will determine whether the foreign gene can be stably expressed during the passage of the recombinant virus.

4. Figure 2B: what is Mock? Explanation in figure legend is needed.

Response: Thanks for your question. The explanation of Mock has been added into the figure legend (line 688) and marked with red color.

5. Figure 2B: while beta-actin provides a control for cell proteins, another control for viral protein(s) would be helpful here.

Response: Thanks for your question. PRV gB protein was used as the viral control and it has been added into the fig 2B.

6. Fig. 4A and 4B: What is the reason of showing results of anti-gB ELISA as signal-to-noise ratio and VP2 ELISA in absorbance units?

Response: Thanks for your question. PRV gB-specific antibody was examine by blocking ELISA, SVA-VP2 specific antibody was examined by the ELISA assay established by our lab. Calculation method of blocking ELISA: $S/N = \frac{\text{the sample } OD_{650nm} \text{ value}}{\text{the negative control } OD_{650nm} \text{ value}}$. The level of PRV antibody in the sample was inversely proportional to the S/N value.

7. Fig.5A is not informative and rather shows raw data. It can be moved into Supplementary materials.

Response: Thanks for your question. So sorry for cannot following this suggestion. Including the Flow cytometry (FCM) scatter plot in the main text can certainly provide a clearer understanding of the experimental results, and it is common practice in many cellular immunity-related articles(8-11). So, in this study, we prefer to put Fig.5A into the main text.

8. Fig. 6A, 6B and 6C replicate data provided in Fig.3. These data can be moved into

Supplementary materials with reference in text that survival kinetic and brain tissue damage as well as serum IL-6 and TNF levels were similar to that found in safety experiment.

Response: Thanks for your question. So sorry for cannot following this suggestion. Fig.3 represents the safety experiment, in which mice were injected with a single dose of the recombinant virus and observed for 14 days. The results demonstrate that rPRV-XJ- Δ TK/gE/gI-VP2 is safe for mice.

In contrast, Fig. 6A, 6B, and 6C depict the PRV challenge experiment. After booster immunization with the recombinant virus, mice were challenged with a virulent strain of PRV at 28 dpv. The results show that rPRV-XJ- Δ TK/gE/gI-VP2 provides 100% protection against PRV infection. Although these figures may appear similar, they represent different groups of mice subjected to different treatments in distinct experiments. Moreover, placing Fig. 6A, 6B, and 6C in the Supplementary materials may impede readers' comprehension of the article.

References

1. Yang JL, Chang CY, Sheng CS, Wang CC, Wang FI. 2020. The Tip Region on VP2 Protein of Bluetongue Virus Contains Potential IL-4-Inducing Amino Acid Peptide Segments. *Pathogens* 10. <https://doi.org/10.3390/pathogens10010003>
2. Li N, Qiao QL, Guo HF, Wang BY, Huang Q, Wang Z, Li YT, Zhao J. 2021. Evaluation of immunogenicity and protective efficacy of a novel Senecavirus A strain-based inactivated vaccine in mice. *Res Vet Sci* 142:133-140. <https://doi.org/10.1016/j.rvsc.2021.12.010>
3. Dall Agnol AM, Miyabe FM, Leme RA, Oliveira TES, Headley SA, Alfieri AA, Alfieri AF. 2018. Quantitative analysis of senecavirus A in tissue samples from naturally infected newborn piglets. *Arch Virol* 163:527-531. <https://doi.org/10.1007/s00705-017-3630-8>
4. Mrak D, Tobudic S, Koblichke M, Graninger M, Radner H, Sieghart D, Hofer P, Perkmann T, Haslacher H, Thalhammer R, Winkler S, Blüml S, Stiasny K, Aberle JH, Smolen JS, Heinz LX, Aletaha D, Bonelli M. 2021. SARS-CoV-2

- vaccination in rituximab-treated patients: B cells promote humoral immune responses in the presence of T-cell-mediated immunity. *Ann Rheum Dis* 80:1345-1350. <https://doi.org/10.1136/annrheumdis-2021-220781>
5. Kawano M, Namba Y, Hanaoka M. 1981. Regulatory factors of lymphocyte-lymphocyte interaction. I. Con A-induced mitogenic factor acts on the late G1 stage of T-cell proliferation. *Microbiol Immunol* 25:505-15. <https://doi.org/10.1111/j.1348-0421.1981.tb00052.x>
 6. Romagnani S. 2000. T-cell subsets (Th1 versus Th2). *Ann Allergy Asthma Immunol* 85:9-18; quiz 18, 21. [https://doi.org/10.1016/s1081-1206\(10\)62426-x](https://doi.org/10.1016/s1081-1206(10)62426-x)
 7. Zhang Z, Zhang Y, Lin X, Chen Z, Wu S. 2019. Development of a novel reverse transcription droplet digital PCR assay for the sensitive detection of Senecavirus A. *Transbound Emerg Dis* 66:517-525. <https://doi.org/10.1111/tbed.13056>
 8. Maggioli MF, Lawson S, de Lima M, Joshi LR, Faccin TC, Bauermann FV, Diel DG. 2018. Adaptive Immune Responses following Senecavirus A Infection in Pigs. *J Virol* 92. <https://doi.org/10.1128/jvi.01717-17>
 9. Yao L, Cheng Y, Wu H, Ghonaim AH, Fan S, Li W, He Q. 2022. The construction and immunogenicity analyses of a recombinant pseudorabies virus with porcine circovirus type 3 capsid protein co-expression. *Vet Microbiol* 264:109283. <https://doi.org/10.1016/j.vetmic.2021.109283>
 10. Zhao J, Zhu L, Xu L, Li F, Deng H, Huang Y, Gu S, Sun X, Zhou Y, Xu Z. 2022. The Construction and Immunogenicity Analyses of Recombinant Pseudorabies Virus With NADC30-Like Porcine Reproductive and Respiratory Syndrome Virus-Like Particles Co-expression. *Front Microbiol* 13:846079. <https://doi.org/10.3389/fmicb.2022.846079>
 11. Blache U, Weiss R, Boldt A, Kapinsky M, Blaudszun AR, Quaiser A, Pohl A, Miloud T, Burgaud M, Vucinic V, Platzbecker U, Sack U, Fricke S, Koehl U. 2021. Advanced Flow Cytometry Assays for Immune Monitoring of CAR-T Cell Applications. *Front Immunol* 12:658314. <https://doi.org/10.3389/fimmu.2021.658314>

March 6, 2023

Prof. Zhiwen Xu
Sichuan Agricultural University
No. 211 , Huimin Road , 11 Wenjiang District, Chengdu City, Sichuan Province , C
Chengdu, Please select
China

Re: Spectrum05229-22R1 (**The construction and immunogenicity analyses of a recombinant pseudorabies virus with Senecavirus A VP2 protein co-expression**)

Dear Prof. Zhiwen Xu:

Your manuscript has been accepted, and I am forwarding it to the ASM Journals Department for publication. You will be notified when your proofs are ready to be viewed.

Sincerely,

Leonidas Stamatatos
Editor, Microbiology Spectrum
